# ACTION SEQUENCE AUGMENTATION FOR ACTION ANTICIPATION

**Yihui Qiu & Deepu Rajan**
College of Computing and Data Science
Nanyang Technological University, Singapore
{qiuy0007,ASDrajan}@ntu.edu.sg

## ABSTRACT

Action anticipation models require an understanding of temporal action patterns and dependencies to predict future actions from previous events. The key challenges arise from the vast number of possible action sequences, given the flexibility in action ordering and the interleaving of multiple goals. Since only a subset of such action sequences are present in action anticipation datasets, there is an inherent ordering bias in them. Another challenge is the presence of noisy input to the models due to erroneous action recognition or other upstream tasks. This paper addresses these challenges by introducing a novel data augmentation strategy that separately augments observed action sequences and next actions. To address biased action ordering, we introduce a grammar induction algorithm that derives a powerful context-free grammar from action sequence data. We also develop an efficient parser to generate plausible next-action candidates beyond the ground truth. For noisy input, we enhance model robustness by randomly deleting or replacing actions in observed sequences. Our experiments on the 50Salads, EGTEA Gaze+, and Epic-Kitchens-100 datasets demonstrate significant performance improvements over existing state-of-the-art methods.

## 1 INTRODUCTION

Action anticipation involves interpreting past events to predict future events in domains such as robotics Antonucci et al. (2021) and surveillance Duque et al. (2007). This capability mirrors human foresight, enabling systems to predict future actions based on observed behaviors, thus allowing for proactive responses.

Predicting next action from an observed action sequence is complex due to three main factors. *Flexibility in the order of actions* allows certain actions to occur in any order, e.g., cut tomatoes, cut cucumber, and cut cheese in ingredient preparation. *Interleaving goals*, where multiple goals are pursued simultaneously, can result in a sequence of potentially uncorrelated actions. For example, in the preparation of coffee and cereal, actions for make coffee, and make cereal can be arranged in multiple valid orders, including in an interleaving manner. As illustrated in Figure 1, the goal of make cereal can occur before the goal of make coffee (as in sequence $a1$), after (as in sequence $a2$), or interleaved with the goal of make coffee (as in sequences $a3$ and $a4$). However, in practice, an action anticipation dataset typically captures only a subset of these possibilities, which can lead to the learning of biased action orders, causing the model to favor specific action sequences and failing to generalize to unseen or less frequent action patterns. Thirdly, action anticipation models often encounter *noisy inputs*, which can arise from misclassified actions, or imprecise temporal annotations, which complicates the task by introducing uncertainty and variability in action prediction.

This paper tackles the above three challenges by generating new action sequences that, when integrated with the original training sequences, broaden the range of possible action orderings, thereby reducing bias and enhancing predictive accuracy. *Sequence augmentation* has been used in natural language processing Hou et al. (2018) and speech recognition Nguyen et al. (2020) to boost model generalization and mitigate over-fitting. Such techniques include synonym replacement, random insertion, deletion, or shuffling of elements. More recently, sequence augmentation has been used in video understanding Prananta et al. (2022); Lee et al. (2022); Falcon et al. (2020), employing

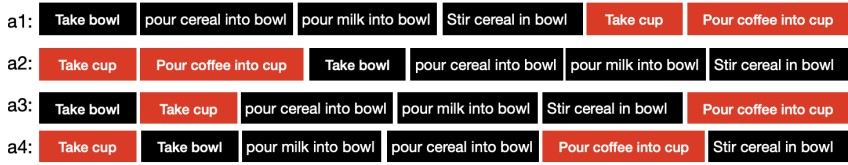

Figure 1: A subset of valid action sequences for make coffee (red) and make cereal(black).

methods like temporal stretching, frame interpolation, and mirroring of video sequences. Despite its success in other domains, augmenting action sequences presents unique challenges that require maintaining logical and temporal coherence to ensure that the augmented sequences remain realistic and meaningful. An effective sequence augmentation strategy for action sequences has yet to be thoroughly investigated.

We introduce a novel sequence augmentation strategy, called $ActSeq$, that separately augments the observed action sequence and the predicted next action. We first propose a novel grammar induction algorithm that extracts a context-free grammar from action sequence labels. This grammar captures the order between actions, guided by the objects and their interactions within the sequence. We also develop a modified version of the Earley Parser Earley (1970) that efficiently generates diverse next action candidates across multiple goals, adhering to the grammar rules and the observed data. Additionally, we enhance the model's robustness by borrowing a proven effective augmentation technique from NLP of random modifications — deleting, or replacing actions — in the observed action sequences. The experiments on three benchmark datasets show that our approach effectively reduces action ordering biases and improves on the state-of-the-art action anticipation models' performances.

## 2 RELATED WORK

**Action Anticipation.** A variety of machine learning models have been explored for action anticipation, each leveraging different aspects of temporal and spatial data processing. Convolutional Neural Networks (CNNs) Abu Farha et al. (2018), long appreciated for their ability to handle spatial features, have been adapted to address the temporal dimensions of video data. Among these adaptations, we focus on RULSTM Furnari & Farinella (2020), which combines recurrent neural network capabilities with CNNs to enhance temporal understanding through state retention over time. Recently, tranformer-based models such as AVT Girdhar & Grauman (2021), RAFTformer Girase et al. (2023) and MotionFormer Patrick et al. (2021) incorporate attention mechanisms that weigh the relevance of different temporal segments. Zhao & Wildes (2020) uses Conditional Adversarial Generative Networks (CAGNs) to jointly anticipate long-term activity labels and their corresponding times.

**Sequence Augmentation** is a methodology that involves artificially expanding the diversity of training datasets by altering existing sequences or synthetically generating new ones, thereby helping models to better generalize across unseen data and mitigate overfitting. Historically, sequence augmentation has been extensively applied in natural language processing (NLP) Li et al. (2022); Feng et al. (2021). Techniques such as synonym replacement Jungiewicz & Smywiński-Pohl (2019), random insertion, deletion, or shuffling of words (or phonemes in speech) Wei & Zou (2019) have been standard practices to inflate text and speech datasets. These methods have proven effective in improving the performance of models on tasks ranging from language translation to speech-to-text recognition. Recently, sequence augmentation has seen innovative applications in video processing Yun et al. (2020); Kwon et al. (2022), such as temporal stretching Prananta et al. (2022), frame interpolation Lee et al. (2022), mirroringFalcon et al. (2020), and generation with Monte-Carlo Tree Search (MTCS) Aziere & Todorovic (2023) of video sequences. To the best of our knowledge, a formal approach to sequence augmentation specifically for action anticipation has not yet been established. Mittal et al. (2024) use a video-language model that enhances the generalization of anticipation models by learning plausible future actions.While large language models (LLMs) excel in scalability and adaptability, they are prone to hallucinations. In contrast, our grammar-based

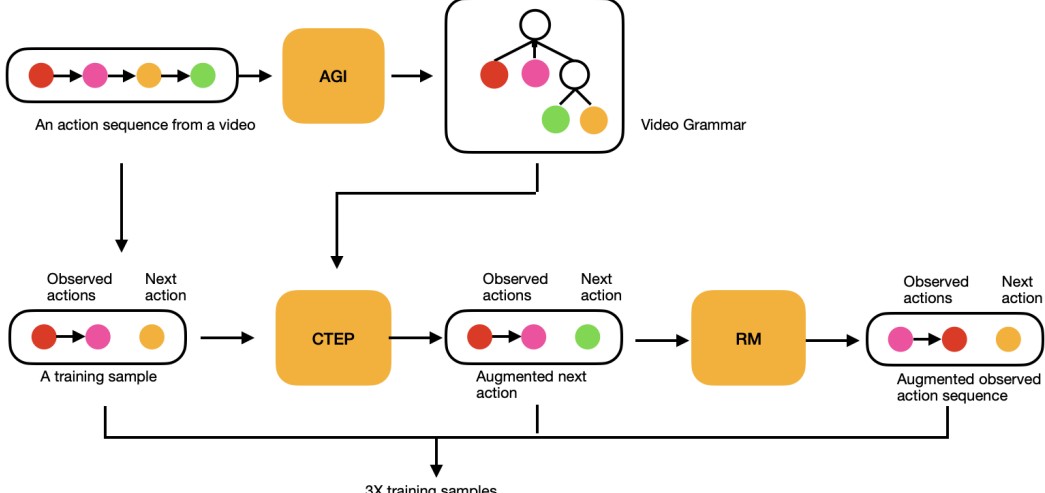

Figure 2: The proposed action sequence augmentation strategy *ActSeq* initiates by deriving video grammar from an action sequence using the Action Grammar Induction (AGI) algorithm. Subsequently, the Cross-Tree Earley Parser (CTEP) predicts alternative next actions based on the observed actions and the video grammar (the figure shows only one augmented sequence). These predicted actions are combined with the original observed actions to compose new training samples. Finally, random modifications (RM) of observed sequences are applied to generate additional training samples. In this example, the training set expands to three times its original size.

method excels in interpretability with explicitly defined and traceable rules. Appendix A compares predictions from representative LLMs and our method.

**Grammar for activity analysis.** Grammar, as a fundamental tool for representing the compositional structure of language, has been extensively explored in the realm of NLP Cremers & Ginsburg (1975); Seki et al. (1991). Beyond textual analysis, grammar's application in action anticipation was first explored by Pei et al. (2011) and Si et al. (2011), who utilized AND-OR grammars learned from strings of symbols, each representing an action according to the grammar's defined language. These early models, however, were limited to deterministic inputs and lacked flexibility. Vo & Bobick (2014) extended this approach by introducing stochastic context-free grammars that could process probabilistic sequence inputs, enhancing the model's adaptability to more realistic, variable data scenarios. More recently, Qi et al. (2020) employed the ADIOS grammar induction algorithm to derive grammars directly from action corpora, marking a significant step towards automation in grammar generation for action sequences. Piergiovanni et al. (2020) proposed a differentiable grammar models that learn sequential dependencies from the action sequence data. Gong et al. (2024) recently introduced a novel grammar induction algorithm that emphasizes key actions and temporal dependencies, incorporating recursive temporal structures to better mirror the repetitive and intertwined nature of human activities. Dessalene et al. (2023) proposed a rule-based, compositional, and hierarchical approach to modeling actions from elemental motions, such as 'grasp' and 'release'. Despite these advancements, many of the existing grammatical frameworks either require hand-crafted rules, or depend on the limited training data that are biased in the order of actions.

## 3    ACTION SEQUENCE AUGMENTATION

Given a video of an action sequence, training samples for action anticipation are typically generated using $M + 1$ continuous actions from the sequence. Here, the first $M$ actions form the observed sequence $a_{\text{obs}} = [a_1, a_2, \ldots, a_M]$, and the $(M + 1)^{\text{th}}$ action is considered as the next action to be predicted, $a_{\text{pred}} = a_{M+1}$.

In sequence augmentation for text classification, e.g., Wei & Zou (2019), augmentation is applied to a training sample $(X, y)$, where $X$ is the input sequence and $y$ is the output class. In our case, $X$ rep-

resents the **observed actions**, and $y$ is the **next action**. Our proposed action sequence augmentation strategy $ActSeq$ applies distinct augmentation techniques to both the observed action sequences $X$ and the next action $y$. The purpose of proposing alternative next actions is to enhance the diversity of potential future actions, aiming to reduce ordering bias. This ensures that the model can accurately predict a variety of possible future actions without depending on a single ground-truth action in the training data. The aim of augmenting observed action sequences is to simulate real-world inaccuracies such as noisy labels in annotation or erroneous action recognition by an action segmentation model. It also forces the model to learn from a broader context of the action sequence under consideration .

The overall pipeline of $ActSeq$ comprises three main steps, as illustrated in Figure 2. **Action Grammar Induction (AGI):** The AGI algorithm initially derives a video grammar from a given action sequence. This grammar serves as the framework for understanding and predicting action dynamics within the video. **Prediction by Cross-Tree Earley Parser (CTEP):** For each training sample extracted from the video sequence, CTEP predicts alternative next actions based on the observed actions and the video grammar. The predicted alternative next actions combine with the original observed actions to form new training samples. In Figure 2, for illustration, we show the CTEP proposing a single next action and, hence, creating one new training sample. **Random Modifications (RM):** For each new training sample generated by CTEP, additional training samples are created through random modifications to the observed action sequences. In Figure 2, for illustration, we show the RM is set to generate one modified observed sequence, hence, creating one additional training sample.

### 3.1 ACTION GRAMMAR INDUCTION (AGI)

We first define two types of **ordering**: dependent and independent. If action $B$ is dependent on action $A$, then $A$ needs to happen before B, denoted by $(A, B)$. On the other hand, if action $A$ and $B$ are independent, they can happen in any order.

As stated in the introduction, learning the correct order of actions is challenging, primarily due to the numerous possibilities of combining actions to construct valid action sequences, particularly when limited by the availability of training samples. However, we observed that actions performed on a single object tend to follow more consistent sequences. For example, in all sequences shown in Fig. 1, actions performed on object cup consistently follow the same order: "take" followed by "pour into." The same pattern holds for objects like the bowl and cereal.

The action grammar is defined as a context-free grammar (Section 3.1.2), designed to derive actions and their orders pertaining to a single object (*atomic action*) or an interaction between multiple objects such as "pour cereal into bowl". Our grammar induction algorithm uses an action dependency matrix, which is derived from the entire training set and maintained as a knowledge base, to construct a unique grammar for each action sequence.

#### 3.1.1 ACTION DEPENDENCY MATRIX

Dependent actions linked to specific objects follow a similar pattern across different goals. Conversely, with independent actions, such as the addition of various ingredients to the bowl, a flexible order of execution is permitted. These aspects are captured in an *action dependency matrix*, which is a skew-symmetric matrix where $M_{(i,j)} = 1$ if action $i$ is dependent on action $j$, and is 0 otherwise.

To construct the matrix, we first isolate the actions associated with each object from all action sequences in the training data, creating object-specific action sequences. These actions could comprise of atomic actions and/or interactions. Interactions among objects are decomposed into respective atomic actions. For instance, for interaction 'pour cereal into bowl', the action for 'cereal' is denoted as 'pour', and for 'bowl' as 'poured into'. Given the object-specifc atomic action sequences, the dependencies between two actions are then computed by thresholding on

$$P_{(A \Rightarrow B)} = \frac{N_{(A \Rightarrow B)}}{N_{(A \Rightarrow B)} + N_{(B \Rightarrow A)}}, \tag{1}$$

where $P_{(A \Rightarrow B)}$ denotes the likelihood that action $B$ is dependent on action $A$, $N_{(A \Rightarrow B)}$ is the number of times action $A$ precedes action $B$ in sequences where both are present. Ideally, if they are independent, $P_{(A \Rightarrow B)} = P_{(B \Rightarrow A)} = 50\%$, and if they are dependent (e.g. $B$ depends on $A$), $P_{(A \Rightarrow B)}$

Table 1: Decomposition of Interactions & Action Dependency Matrix. Interactions are decomposed into atomic actions and their dependencies. $M(i,j) = 1$ : action $i$ is dependent on action $j$, $M(i,j) = -1$: reverse dependency, $M(i,j) = 0$ (shown as -): no dependency between actions.

| Interaction | Atomic Actions | Take bowl | Pour cereal | Poured into bowl | Pour milk | Poured into bowl | Stir cereal | Stir in bowl |
|---|---|---|---|---|---|---|---|---|
| | Take bowl | - | - | -1 | - | -1 | - | - |
| Pour cereal into bowl | Pour cereal | - | - | - | - | - | -1 | - |
| | Poured into bowl | 1 | - | - | - | - | - | -1 |
| Pour milk into bowl | Pour milk | - | - | - | - | - | - | - |
| | Poured into bowl | 1 | - | - | - | - | - | -1 |
| Stir cereal in bowl | Stir cereal | - | 1 | - | - | - | - | - |
| | Stir in bowl | - | - | 1 | - | 1 | - | - |

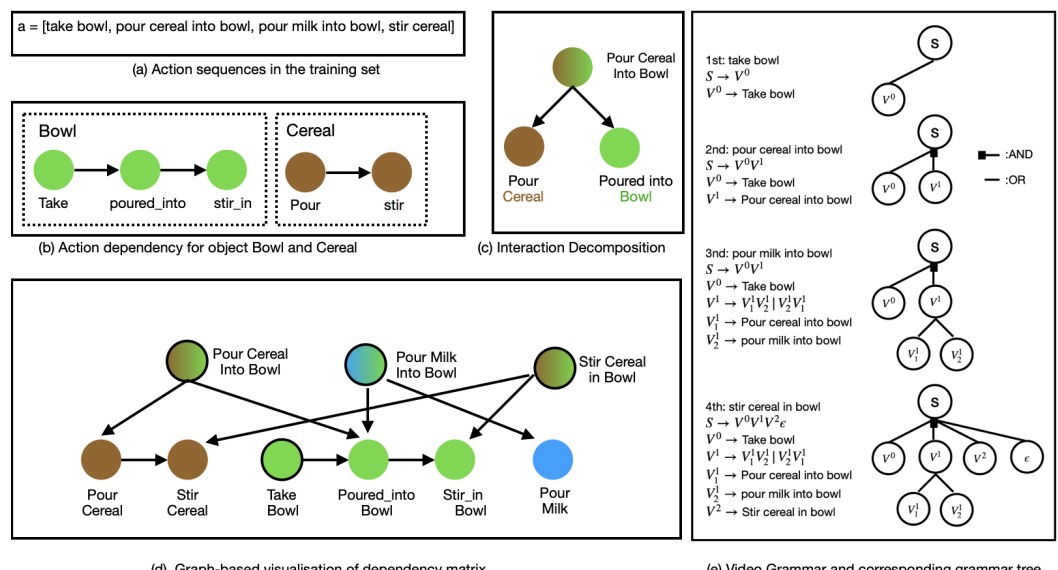

Figure 3: Example of Video Grammar construction. From an action sequence (a), action dependencies of atomic actions for each object (b), and interaction decomposition (c) of all atomic actions and interactions in the sequence, are used to construct a dependency matrix for the sequence, visualized as a graph (d). (e) demonstrates step by step construction of video grammar's production rules and corresponding grammar tree from the dependency matrix.

should be 100%. However in real datasets, this is usually not true due to the bias caused by limited samples and incorrect or missing labels. Hence, we introduce a threshold, if $P_{(A \Rightarrow B)}$ is greater than this threshold, action $B$ is deemed dependent on action $A$ and $M_{(A,B)} = 1$. Otherwise the two actions are considered independent and $M_{(A,B)} = 0$. The action dependency matrix that captures the sequential dependencies of actions is shown in Table1.

Figure 3 (b) provides a graph-based visualization of atomic action dependencies for objects. In (c), the graph shows the decomposition of interaction into atomic actions. Finally, (d) presents a graphical representation of the action dependencies matrix in Table.1for action sequence in (a).

### 3.1.2 VIDEO GRAMMAR

Formally, a *context-free grammar* (CFG) Cremers & Ginsburg (1975) is defined as a 4-tuple $G = (C, T, P, S)$, where $C$ is a finite set of non-terminal symbols, representing higher-level constructs. $T$ is a finite set of terminal symbols. $P$ is a finite set of production rules of the form $A \rightarrow \alpha$, where $A \in C$ is a non-terminal and $\alpha \in (C \cup T)^*$. The $*$ represents the set of all strings that composed of non-terminals and terminals. The $\rightarrow$ denotes replacement of the non-terminal $A$ with the sequence $\alpha$. $S$ is the start symbol.

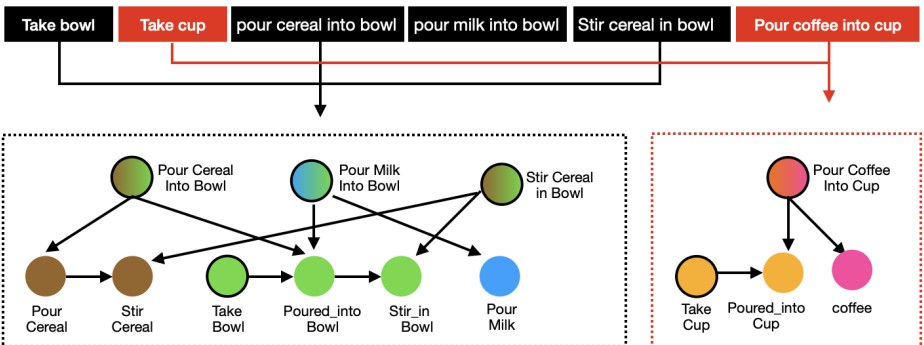

Figure 4: Given an action sequence with two entangled goals, AGI separates the two goals and represents them using two dependency matrices, visualised as two graphs.

We use two types of production rules, 'AND' and 'OR', defined as (i) **AND** : $C \rightarrow \alpha\beta$, and (ii) **OR** : $C \rightarrow \alpha\beta \mid \beta\alpha$. The **AND** rule dictates that the non-terminal $C$ is replaced by the sequence $\alpha\beta$, establishing that $\alpha$ occurs before $\beta$. Conversely, the **OR** rule, containing the '|' operator, allows for the non-terminal $C$ to be replaced by either $\alpha\beta$ or $\beta\alpha$, signifying that $\alpha$ and $\beta$ can occur in any order. These rules enable the hierarchical generation of action sequences. The video grammar is constructed using the dependency matrix. Actions that are dependent are linked with an **AND** operator, while independent actions are connected using an **OR** operator.

We demonstrate the construction of the video grammar with the example shown in Figure 3, starting with the initial action "Take bowl." As this action does not depend on any others, it is directly added to the root node $S$ as $V^0$ in Figure 3(e). The next action "Pour cereal into bowl" relies on "Take bowl", because its component action "poured into bowl" is dependent on "Take bowl". Consequently, it is connected to the right of $V^0$ at the root node as $V^1$, using an **AND** operator to indicate that $V^0$ precedes $V^1$. The third action, "Pour milk into bowl," also depends on "Take bowl" but is independent of "Pour cereal into bowl." To manage this independence, a dummy node $V^1$ replaces the original $V^1$, now termed $V_1^1$, representing "Pour cereal into bowl." The action "Pour milk into bowl" is then introduced as $V_2^1$ and connected to $V^1$ with an **OR** operator, indicating that $V_1^1$ and $V_2^1$ can occur in any order. Lastly, "Stir cereal into bowl" is added as $V^2$ to the root node because it depends on both $V_1^1$ and $V_2^1$. Figure 3(e) illustrates the fully constructed video grammar's production rules and corresponding grammar tree.

The example demonstrates that the proposed AGI algorithm is both simple and effective at capturing correct action ordering. Another advantage is that when there are multiple goals being pursued at the same time, for instance, "make cereal" and "make coffee", AGI is able to separate the two goals and represent them using two dependency matrices, visualised as a graph in Figure 4.

## 3.2 CROSS-TREE EARLEY PARSER

The primary objective of the parser is to propose next action candidates based on observed actions, using the video grammar as a guide. When multiple grammar trees are present, the original Earley Parser will not predict actions from a grammar tree if none of its actions has been observed. For instance, as illustrated in Figure 4, if the observed input is "Take bowl," belonging to the "make cereal" grammar tree, the Earley Parser will not predict "Take cup," which is part of the "make coffee" grammar tree, Instead, it will continue to process actions only from the "make cereal" grammar tree.

We propose a modified version of the Earley parser called Cross-Tree Earley parser (CTEP) that can handle interleaving actions from multiple goals. It operates similarly to the Earley Parser but includes enhancements to handle goal switching and parallel goal execution. However, allowing parsing across multiple trees will exponentially expand the search space as the number of trees in a video increases. To enhance search efficiency, the CTEP Parser allows selective initialization of grammar trees. In our experiments, only the trees containing observed actions and the next $N$ actions are initialized.

We first add an initialization process to allow initializing multiple grammar trees. A terminal symbol ">" is added as the left child of the root node of selected grammar trees; it also serves as the first input to initialize these grammars. After initialization, the CTEP parser maintains a set of states at each position in the input sequence of observed actions, similar to the Earley Parser. As it moves through the input, it updates the set of states according to three primary operations - Prediction, Scanning and Completion. In the following, the symbols $P$ and $Q$ refer to the non-terminals. The symbols $\alpha$ and $\beta$ represent arbitrary strings consisting of terminals and non-terminals. The dot ($\cdot$) denotes the current position of the parser within the production rule.

(i) **Prediction:** If the current state expects a non-terminal ($P \rightarrow \alpha \cdot Q\beta$), the parser outputs all possible productions of $Q$ that can be executed. If the current state expects a terminal, the parser moves on to the Scanning phase. When all the actions in the input are processed, the parser outputs all possible next terminals in the current states as the candidates for next action. The candidates are then prioritized based on the order in which they appear in the video sequence. This prioritization reflects the natural progression of events in the video. (ii) **Scanning:** The parser checks the next input symbol in the current state. If it matches the expected terminal, the parser moves the dot to the right, indicating successful scanning of that part of the input. (iii) **Completion:** When the dot reaches the end of a production ($P \rightarrow \alpha\beta\cdot$), the parser "completes" that production. The parser then revisits previous states that were anticipating this non-terminal $P$ and advances the dot in those states.

### 3.3 RANDOM MODIFICATION ON OBSERVED ACTIONS

Inspired by the effective text classification technique of random insertion, deletion, and replacement of words, which robustly enhances training sample diversity and improves model generalization Wei & Zou (2019), we have adopted a similar augmentation strategy for action sequences. For each sequence in the training set, we randomly apply one of the following operations to $n = \lfloor L/k \rfloor + 1$ actions, where $L$ represents the total number of actions in a training sample and parameter $k$ is an adjustable impact factor that controls the number of actions to be modified. **Action Replacement (AR):** Randomly select actions from the sequence and replace each of these actions with one chosen at random from the sequence. This simulates scenarios where actions may be misclassified by upstream models or by wrong labeling in the annotation. **Action Deletion (AD):** Remove randomly selected action(s) in the sequence. This simulates scenarios where actions may be missed or unobserved during recording or recognition.

## 4 EXPERIMENTS

### 4.1 DATASETS

We perform experiments on three action anticipation benchmarks. **50Salads** Stein & McKenna (2013) consists of 901 action annotations, and 17 action classes. We report the average performance across the standard five splits. **EGTEA Gaze+** Li et al. (2021) contains 10,325 action annotations, 19 verbs, 51 nouns and 106 action classes. Methods are evaluated on EGTEA Gaze+ reporting the average performance across the three splits provided by the authors of the dataset. **EPIC-Kitchens-100** Damen et al. (2022) contains 3806 actions, with 97 verbs, and 300 nouns. Action is represented in (verb, noun) pair. For example, action "open fridge" consists of verb "open", and noun "fridge". We evaluate our method on the validation dataset following previous work Guo et al. (2024).

### 4.2 METRICS

We use Mean Top-5 Recall as a class aware metric for Epic-Kitchens and EGTEA Gaze+ following Furnari et al. (2019). Specifically, when assessing performance on EGTEA Gaze+, the Top-5 Recall is calculated by averaging across actions. In contrast, for EPIC-Kitchens, the metric averages over verbs, nouns, and actions. For the 50Salads dataset, which contains significantly fewer actions, we use Top-1 accuracy as the evaluation metric, following Guo et al. (2024).

We also propose a metric called Mean Action Order Bias (MAOB) to measure the bias in action order within a dataset. Specifically, for each independent pair of actions, $A$ and $B$, the action order bias for a pair is defined as $AOB_{AB} = |P_{(A \Rightarrow B)} - P_{(B \Rightarrow A)}|$, where $P_{(A \Rightarrow B)}$ is the probability that

$A$ occurs before $B$. These probabilities are computed from training samples where both $A$ and $B$ are present. The Mean AOB is then computed by averaging over all action pairs.

### 4.3 STUDY OF ACTION ORDER BIAS IN 50SALADS DATASET

Table 2 compares the MAOB for 50Salads dataset between the original sequence and with the augmented sequence generated from $ActSeq$. The MAOB of the original sequence is 0.32, indicating the presence of significant bias. The main source of bias arises from ingredients preparation (e.g., cutting various ingredients and placing them in a bowl) and from sauce preparation (e.g., combining different sauces in a cup) processes. Both processes consist of 4 to 5 independent actions, but some of these actions are significantly more likely to occur in a particular order. For instance, in 67% of the videos, the action "cut tomato" occurs before "cut cucumber," and in 71% of the videos, "add oil" precedes "put salt". After incorporating our augmented sequences into the training samples, the MAOB significantly decreased to 0.12, demonstrating the effectiveness of our method in reducing ordering bias. MAOB studies on EGTEA Gaze+ and Epic-Kitchens-100 are in Appendix B.

Table 2: Mean Action Order Bias on 50Salads.

| Method | MAOB |
|---|---|
| Original sequence | 0.32 |
| Original + augmented sequence by $ActSeq$ (ours) | **0.12** |

### 4.4 ABLATION STUDY OF ACTSEQ ON EGTEA GAZE+ DATASET

In this section, we study the impact of augmenting observed actions and next action individually, as well as in combination on the performance of action anticipation on the EGTEA Gaze+ dataset. We also compare our augmentation strategy with other common sequence augmentation methods using a state-of-the-art model based on RAFTformer Girase et al. (2023). Ablation study on 50Salads and Epic-Kitchens-100 can be found in Appendix C.

**Augmentation to observed actions and next action**. Table 3 and Table 4 show RAFTFormer's Mean Top 5 Recall on EGTEA Gaze+ changes as a function of two variables: (i) the size of augmented sequences by modifying the observed action sequences and (ii) the size of augmented sequences by modifying the next action—expressed as multiples of the original training sequences. For example, "3X" under Observed in Table 3 indicates that the number of new sequences generated by augmenting observed actions is three times the size of the original set.

Table 3: Ablation on number of augmented observed actions on EGTEA Gaze+.

| Observed | Next | Top 5 Recall |
|---|---|---|
| original | | 62.1 |
| 2X | | 62.8 |
| 3X | original | **63.1** |
| 4X | | 63.0 |
| 5X | | 62.5 |

Table 4: Ablation on number of augmented next actions on EGTEA Gaze+.

| Observed | Next | Top 5 Recall |
|---|---|---|
| | original | 62.1 |
| | 1X | 63.5 |
| original | 2X | 64.7 |
| | 4X | 65.2 |
| | 5X | **65.3** |

Table 5: Ablation on number of augmented observed and next actions on EGTEA Gaze+.

| Observed | Next | Top 5 Recall |
|---|---|---|
| original | original | 62.1 |
| 1X | 1X | 63.8 |
| 2X | 2X | 65.2 |
| 3X | 4X | **66.3** |
| 4X | 5X | 65.5 |

Table 3 shows the augmentation effects on observed action sequences. The recall rate initially improves from 62.1% with no augmentation to 63.1% at a 3X increase, suggesting that moderate augmentation to the observed action sequence positively impacts model training by forcing the model to

learn from the context instead of relying on a few key actions. However, further increase to 4X and 5X augmentation leads to a decrease in performance, dropping to 62.5%. This indicates a potential over-modification of the observed action sequences, where learning next action from the augmented sequence becomes too difficult. The subtle fluctuation in recall rates across augmentation levels suggests that while augmentation helps to a certain extent, its benefits are non-linear and can lead to negative returns when excessive.

On the other hand, Table 4 shows a slightly different trend when augmenting next actions. The recall rates steadily increase from 62.1% without augmentation to a peak of 65.3% at 5X augmentation, though with diminishing returns. Notably, 5X is the maximum number of candidates available for the ETGEA Gaze+ dataset. This indicates a consistent improvement as augmenting next actions introduces valuable predictive complexity, helping the model better anticipate and generalize future actions.

Table 5 provides an integrated view by comparing Top 5 Recall when both observed and next actions are augmented. For example, 3X under Observed and 4X under Next means that there are three times the augmented observed actions for each of the four augmented next actions. In this case, the recall peaks at 66.3%, which demonstrates that combining augmentation techniques on both observed and next actions yields a synergistic effect, outperforming the application of either method in isolation. Additionally, the results suggest that there is an optimal scaling factor for the observed and next action sequences that maximizes performance.

**Comparison with other sequence augmentation methods**. Since sequence augmentation methods have not been previously applied to action anticipation, we adopt two commonly used methods from other fields. Method 1 is from text classification, which involves random modifications (deletion, insertion, addition, or replacement) of words in text Wei & Zou (2019). Method 2 is from multi-label image classification, which utilizes prediction results from a downstream model Ke et al. (2019). For method 1, we apply modifications to randomly selected actions as detailed in Section 3.3 to the video sequence, then generate training samples from the augmented sequence. For method 2, we take the predicted next actions from AVT Girdhar & Grauman (2021), combine each prediction with the observed actions from the original training samples to create augmented samples. Additionally, we propose a new approach, Method 3, called "NEXT D." This method uses each of the next D actions following the observation as the "next action", combined with the original observed actions to generate D training sequences. For example, consider the notation $[a_{obs}, a_{pred}] = [[a, b, c], d]$ where $[a, b, c]$ represents the observed actions and $d$ is the next action. If the action sequence in a video is $[a, b, c, d, e, f, g, ...]$, and a training sample is $[[a, b, c], d]$, then the "NEXT D" augmented sequences with $D = 2$ are $[[a, b, c], e]$ and $[[a, b, c], f]$.

Table 6: Comparison of RAFTformer's performance trained on other augmentation methods on EGTEA Gaze+ dataset. $a_{obs}$ is the observed sequence, $a_{pred}$ is the next action.

| Method | Applied to | Recall (%) |
|---|---|---|
| Original training set | - | 63.5 |
| Original + Randomly delete, insert, add, replace actions (5X) | $a_{obs}, a_{pred}$ | 62.2 |
| Original + Predicted results by AVT (5X) | $a_{pred}$ | 63.6 |
| Original + Next 5 (5X) | $a_{pred}$ | 64.8 |
| Original + ActSeq (pred only) (5X) (Ours) | $a_{pred}$ | **65.3** |
| Original + ActSeq (12X) (Ours) | $a_{obs}, a_{pred}$ | **66.3** |

Table 6 shows RAFTformer's performance with augmented sequences generated by different methods on the EGTEA Gaze+ dataset. The application of randomly deleting, inserting, and replacing actions to the whole video sequence (i.e.,both observed and next actions) leads to a decrease in performance. In contrast, as noted in Table 3, augmentation applied solely to observed actions resulted in performance improvements. This suggests that random modifications of next actions introduce significant noise or irrelevant changes, leading to worse performance. Additionally, using predictions from AVT slightly enhances performance implying that while different models may learn distinct insights from the dataset, they may yield less diversity and accuracy of possible next actions. Interestingly, the simpler "Next D" method significantly boosts performance to 64.8% when $D = 5$, indicating a strong correlation between observed actions and the next few actions. In comparison, $ActSeq$, when applied on next action only, achieves a higher recall of 65.3%. $ActSeq$ is able to

generate valid action sequences even when the size of augmented sequences is 12 times the original (i.e. 12X) achieving the best recall of 66.3%.

## 4.5 IMPACT OF SEQUENCE AUGMENTATION STRATEGY ON SOTA

In this experiment, we evaluate state-of-the-art models trained with and without $ActSeq$ on 50Salads, EGTEA Gaze+, and EPIC-Kitchens-100 datasets, each offering varying complexity in terms of action classes, goal diversity, and video length. This allows us to assess our method's generalizability across different models and datasets.

The 50Salads dataset (Table.7), with fewer action classes, a single goal and shorter sequences, results in relatively fewer action ordering possibilities. Even so, reducing ordering bias through ActSeq improves the performance of all models, albeit by a small amount. In contrast, EGTEA Gaze+ (Table.8) and EPIC-Kitchens-100 (Table.9) present a much more complex action space with diverse verbs and nouns across multiple goals, where ActSeq significantly improves recall by enabling models to capture subtle action ordering variations.

Appendix D contains a step by step demonstration of how our ActSeq method generates augmented training samples using an example from the Epic-Kitchens dataset. Appendix E demonstrates the model's efficiency when trained with low-data.

In summary, these experiments highlight ActSeq's robustness across different models and benchmarks. Its consistent performance gains demonstrate its potential as a valuable strategy for improving action anticipation for datasets with different complexities.

Table 7: Top 1 Accuracy on 50Salads.

| Method | Action |
|---|---|
| CNNAbu Farha et al. (2018) | 29.8 |
| CNN + ActSeq | **31.1** |
| AVTGirdhar & Grauman (2021) | 48.0 |
| AVT + ActSeq | **49.7** |
| RAFTformerGirase et al. (2023) | 53.2 |
| RAFTformer + ActSeq | **53.9** |

Table 8: Top 5 recall on EGTEA Gaze+.

| Method | Action |
|---|---|
| RULSTMFurnari & Farinella (2020) | 58.6 |
| RULSTM + ActSeq | **60.4** |
| AVTGirdhar & Grauman (2021) | 62.1 |
| AVT + ActSeq | **64.8** |
| RAFTformerGirase et al. (2023) | 63.5 |
| RAFTformer + ActSeq | **66.3** |

Table 9: Top 5 recall metrics on the Epic-Kitchen-100 dataset.

| Method | Verb | Noun | Action |
|---|---|---|---|
| RULSTMFurnari & Farinella (2020) | 30.8 | 27.8 | 14 |
| RULSTM + ActSeq | **32.1** | **30.3** | **15.9** |
| RAFTformerGirase et al. (2023) | 33.8 | 37.9 | 19.1 |
| RAFTformer + ActSeq | **35.1** | **40.4** | **20.8** |
| MotionFormerPatrick et al. (2021) | 47.1 | 46.9 | 21.5 |
| MotionFormer + ActSeq | **48.4** | **49.4** | **22.4** |

## 5 CONCLUSION AND FUTURE WORK

This paper introduces a comprehensive framework aimed at mitigating the challenges of action ordering bias in datasets and noisy input to action anticipation models through innovative sequence augmentation techniques. By tailoring distinct augmentation methods to the observed actions and the next action, we effectively reduce biases in action order and improve the model's resilience to noisy input. This approach has demonstrated significant performance improvements across various datasets and state-of-the-art models.

However, our current framework assumes that when multiple goals are pursued simultaneously, the action sets for each goal are independent. This simplification may not hold in cases where a single action contributes to multiple goals. Future research could extend our model to address such complexities by exploring how intertwined action sequences and overlapping goals can be more effectively represented and anticipated.

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

## A    COMPARING GRAMMAR-BASED METHODS AND LLMS FOR ACTION SEQUENCE AUGMENTATION IN ACTION ANTICIPATION

The integration of grammar-based methods in action sequence augmentation has demonstrated promising results due to their interpretability and structured approach to generating action sequences. However, Large Language Models (LLMs) such as ChatGPT also offer compelling alternatives due to their capability to generate diverse and contextually rich action sequences from extensive training data. Comparing these methodologies is essential for understanding their respective strengths and limitations in various application scenarios, particularly in dynamic and less-structured environments.

### A.1    EXPERIMENTAL SETTING

- **Dataset: 50Salads**
- **Task:** Predict legitimate next actions given a predefined sequence of cooking actions.
- **Models Evaluated:**
    - **Proposed Grammar-based Method (ActSeq):** Utilizes rules derived from action narrations to generate sequences.
    - **Gemini 1.5 Flash, Claude, ChatGPT-3.5 and ChatGPT-4.0:** These models represent LLMs, chosen for their varying capabilities in understanding and generating text-based sequences.

### A.2    EXAMPLES AND COMPARISON OF RESULTS

- **Prompt given to LLM models:**
    - **Task:** Given the complete action sequence, the observed actions and the goal, determine what the next action could be (if there are multiple plausible actions, list them all). Provide a justification for your choice.
    - **Complete Action Sequence:**[cut tomato, place tomato into bowl, cut cheese, place cheese into bowl, cut lettuce, place lettuce into bowl, add salt, add vinegar, add oil, add pepper]
    - **Observed Actions:**[cut tomato, place tomato into bowl]
    - **Goal:** Prepare a mixed salad with dressed vegetables.
- **Predictions and Analysis:**
    - **Gemini 1.5 Flash:**
        * **Predicted Actions:** 'cut cheese'

* **Justification:** The goal is to prepare a mixed salad with dressed vegetables. The observed actions indicate that the tomato has been prepared and added to the bowl. The next logical step in preparing a mixed salad is to prepare another ingredient, such as the cheese. The action sequence shows that cutting and placing the cheese into the bowl follows the preparation of the tomato. Therefore, cutting cheese is the most plausible next action based on the goal and the observed actions.
* **Analysis:** The model correctly states that "The next logical step in preparing a mixed salad is to prepare another ingredient." However missed 'cut lettuce' in its prediction. Its prediction could be copied from the original action sequence.

– **Claude:**
* **Predicted Actions:** Cut cheese and place it into the bowl or Cut lettuce and place it into the bowl.
* **Justifications:** The goal is to prepare a mixed salad, and the complete action sequence indicates that the next step after placing the tomato into the bowl is to cut the cheese and place it into the bowl, or cut the lettuce and place it into the bowl. This aligns with the goal of creating a mixed salad with various ingredients.
* **Analysis:** The model seems to interpret "Cut cheese and place it into the bowl" as a single action, or a immediate sequence of actions, suggesting that once the cheese is cut, it should directly be placed in the bowl. However, there might be instances where multiple ingredients are cut, then put into bowl afterwards.

– **ChatGPT-3.5:**
* **Predicted Actions:** 'cut cheese', 'cut lettuce', 'add salt'
* **Justifications:** cut cheese and cut lettuce are independent of previous steps beyond what's observed. Add salt could technically happen after placing the tomato in the bowl.
* **Analysis:** The model suggests adding salt immediately after the initial salad preparation steps, showcasing a lack of understanding of the structured sequence typically followed in salad preparation.

– **ChatGPT-4o:**
* **Predicted Actions:** 'cut cheese', 'cut lettuce'
* **Justifications:** The most plausible next action is "cut cheese." Alternatively, "cut lettuce" could also be a valid action. Both align with the goal of preparing a mixed salad by incorporating the next ingredient into the preparation process.
* **Analysis:** The model correctly predicts the two candidates, however in its justification, it prefers "cut cheese" over "cut lettuce. it could due to "cut cheese" is the one that appears in the ground truth action sequence.

– **ActSeq (Grammar-based Method):**
* **Predicted Actions:** 'cut cheese', 'cut lettuce'
* **Analysis:** Correctly identifies the need to continue with ingredient preparation before moving on to dressing components, demonstrating strong adherence to the structured task rules.

The analysis of Large Language Models (LLMs) in the action sequence augmentation experiment reveals several weaknesses, including varied performance based on model capabilities, inherent biases, and occasional misunderstandings of task. While more sophisticated prompt engineering can improve LLM responses by guiding them towards more accurate predictions, it remains challenging to anticipate and cover all potential scenarios.

# B EXTENDED MAOB STUDY ON EGTEA GAZE+ AND EPIC-KITCHENS-100 DATASETS

To evaluate the generalizability of our ActSeq method, we extended the MAOB study to the EGTEA Gaze+ and Epic-Kitchens-100 datasets in addition to 50Salads. The purpose of this study is to assess how well ActSeq mitigates action order biases in datasets with varying levels of complexity. For reference, 50Salads contains fewer action classes, a single goal, and shorter video sequences,

while EGTEA Gaze+ and Epic-Kitchens-100 feature a more diverse set of verbs, nouns, and goals, presenting a significantly more complex action space.

As shown in Table 10, the augmented sequences generated by our ActSeq method are 12 times (3X for observed action and 4X for predicted action) of the original sequences. ActSeq consistently reduces MAOB across all three datasets, significantly lowering the bias from 0.32 to 0.12 in 50Salads, from 0.28 to 0.10 in EGTEA Gaze+, and from 0.46 to 0.19 in Epic-Kitchens-100. This demonstrates that ActSeq effectively mitigates action order biases, resulting in more balanced and representative datasets.

Notably, the reduction in MAOB correlates with improved performance across all datasets. By minimizing biases in action ordering, ActSeq enables models to better generalize to unseen sequences, especially in datasets like EGTEA Gaze+ and Epic-Kitchens-100, where subtle variations in action order can significantly impact performance. These findings further highlight the robustness of ActSeq and its potential for generalization across diverse datasets.

Table 10: Mean Action Order Bias (MAOB) across datasets.

| Method | 50Salads | EGTEA Gaze+ | Epic-Kitchens-100 |
|---|---|---|---|
| Original sequence | 0.32 | 0.28 | 0.46 |
| Original + $ActSeq$ (ours) | **0.12** | **0.10** | **0.19** |

The extended MAOB study provides strong supporting evidence for the effectiveness of ActSeq in reducing action order biases across datasets with varying complexity. These results align with observed performance improvements, emphasizing the robustness and generalizability of our method. This analysis also underscores the importance of addressing biases to enhance model performance in diverse and challenging action understanding tasks.

## C   ABLATION STUDY OF AUGMENTATION RATIO ON 50SALADS AND EPIC-KITCHENS-100 DATASETS

This appendix conducts a similar ablation study as in Tables 3 and 4 for the 50Salads and Epic-Kitchens-100 datasets. These datasets differ from EGTEA Gaze+ in size, domain, and data distribution. The additional studies aim to evaluate the generalizability of the ActSeq method across datasets with varying characteristics.

The results of the ablation studies are summarized below for both datasets. These studies assess the impact of varying augmentation ratios for observed and next action sequences on Top-5 Recall performance.

**50Salads Dataset:** The 50Salads dataset, characterized by fewer action classes, shorter sequences, and a single goal, showed consistent improvements with ActSeq. As seen in Table 11, the best performance was observed with a configuration of 2X observed and 3X next actions.

**Epic-Kitchens-100 Dataset:** The Epic-Kitchens-100 dataset, with its complex action space and diverse goals, showed steady improvements in Top-5 Recall with increased augmentation ratios (Table 12). The best performance of 20.8% achieved at a 3X observed and 4X next configuration. These results further support the effectiveness of ActSeq in handling diverse and complex datasets.

The ablation studies on 50Salads and Epic-Kitchens-100 confirm the generalizability of ActSeq to datasets of varying size, domain, and complexity. Moderate augmentation levels (e.g., 2X or 3X) strike a balance between computational efficiency and performance improvement, making ActSeq a practical and adaptable solution for enhancing model performance in diverse applications.

## D   DEMONSTRATION EXAMPLE FROM THE EPIC-KITCHENS DATASET

This appendix demonstrates how our ActSeq method generates augmented training samples from the Epic-Kitchens dataset. The example illustrates the derivation of action dependencies, their representation in a grammar tree, and the step-by-step generation of augmented sequences using our approach.

Table 11: Ablation study on the 50Salads dataset: Top-5 Recall (%).

| Observed | Next | Top-5 Recall (%) |
|---|---|---|
| original | original | 53.2 |
| 1X | original | 53.4 |
| 2X | original | **53.5** |
| 3X | original | 53.3 |
| 4X | original | 53.1 |
| original | 1X | 53.5 |
| original | 2X | 53.6 |
| original | 3X | **53.7** |
| original | 4X | **53.7** |
| 1X | 1X | 53.5 |
| 2X | 2X | 53.7 |
| 2X | 3X | **53.9** |
| 2X | 4X | 53.5 |

Table 12: Ablation study on the Epic-Kitchens-100 dataset: Top-5 Recall (%).

| Observed | Next | Top-5 Recall (%) |
|---|---|---|
| original | original | 19.1 |
| 2X | original | 19.4 |
| 3X | original | 19.5 |
| 4X | original | **19.6** |
| 5X | original | 19.5 |
| original | 1X | 19.8 |
| original | 2X | 20.2 |
| original | 4X | **20.5** |
| original | 5X | **20.5** |
| 1X | 1X | 19.9 |
| 2X | 2X | 20.3 |
| 2X | 4X | 20.6 |
| 3X | 4X | **20.8** |

ORIGINAL SEQUENCE AND DEPENDENCIES

The original sequence of actions is as follows:

**Original Sequence:** [open drawer, take knife from drawer, put knife on table, take pot from drawer, close drawer, put pot on stove.]

Using object-based action transitions and interaction decomposition, the following dependencies are derived:

- open drawer → close drawer
- open drawer → take knife from drawer
- open drawer → take pot from drawer
- take knife from drawer → put knife on table
- take pot from drawer → put pot on stove

These dependencies can be represented as a grammar tree, as shown in Figure 5.

TRAINING SAMPLE EXTRACTION

From the original sequence, training samples are extracted by considering fixed-length observed action sequences. In this demonstration, we assume a length of two actions for the observed sequence. This results in the following training samples, each in the format: [[observed actions], next action]

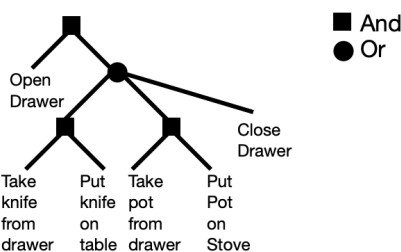

Figure 5: Grammar tree representation of the action sequence: open drawer, take knife from drawer, put knife on table, take pot from drawer, close drawer, put pot on stove.

1. [[open drawer, take knife from drawer], put knife on table]
2. [[take knife from drawer, put knife on table], take pot from drawer]
3. [[put knife on table, take pot from drawer], close drawer]
4. [[take pot from drawer, close drawer], put pot on stove]

AUGMENTATION EXAMPLE

We demonstrate the augmentation process using the first training sample:

[[observed actions], GT next action] = [[open drawer, take knife from drawer], put knife on table]

**Predicting Next Action Candidates**    Using CTEP (our candidate generation method), the following **next action candidates** are predicted based on the observed actions:

{put knife on table, take pot from drawer, close drawer}.

**Generating Augmented Sequences**    To generate augmented sequences, we exclude the ground truth (GT) next action (*put knife on table*) and combine the observed actions with other plausible candidates. The resulting augmented sequences are:

a.    [[open drawer, take knife from drawer], take pot from drawer]
b.    [[open drawer, take knife from drawer], close drawer]

**Applying Random Modifications (RMs)**    To enhance diversity, random modifications (RMs) are applied to the observed actions. For example, switching the order of two actions produces:

c.    [[take knife from drawer, open drawer], take pot from drawer]
d.    [[take knife from drawer, open drawer], close drawer]

In this example, a total of 4 augmented sequences (a,b,c, and d) are generated. This example demonstrates how ActSeq generates diverse augmented training samples. By combining plausible next actions with observed sequences and applying random modifications, our method effectively enriches the training data while adhering to the dependencies derived from the grammar tree.

# E    DEMONSTRATION OF MODEL EFFECTIVENESS IN LOW-DATA MODE

This appendix demonstrates how our approach improves data efficiency of a network, enabling it to require fewer original action sequences during training.

EXPERIMENTAL SETUP

We simulated low-data conditions by selecting a limited number of training videos per task from the EGTEA Gaze+ dataset. Specifically, we created subsets with 3 and 5 videos per task. These subsets were used to train the model both with and without ActSeq augmentations. The performance of these low-data models was compared to that of a full-data model trained with an average of 10 videos per task.

The RAFTFormer model was used as the base, and Top-5 Recall was selected as the evaluation metric. Table 13 summarizes the results.

Table 13: Performance of the low-data model on EGTEA Gaze+ with ActSeq.

| Videos per Task | With ActSeq (Top-5 Recall) |
|---|---|
| 3 | 64.7 |
| 5 | 65.8 |
| Full (on average 10 per task) | 66.3 |

RESULTS AND DISCUSSION

Table 13 demonstrates the consistent improvement in performance achieved with ActSeq across varying levels of training data availability. With only 3 videos per task, the model achieves a Top-5 Recall of 64.7%. This result highlights the robustness of ActSeq in extremely limited data settings, where it provides sufficient diversity to improve learning outcomes. It's worth noting that this performance is surpassing the RAFTFormer model trained with full original training set, which is 63.5%.

As the number of training videos per task increases to 5, the Top-5 Recall improves to 65.8%, showing that it enables models to achieve performance levels close to those of full-data models (66.3%). This result illustrates that while larger datasets naturally lead to better performance, ActSeq ensures that models trained on smaller datasets can still perform competitively.

The experiments validate the data efficiency of ActSeq, showing that it enables models to achieve performance levels close to those of full-data models, even with fewer available original training samples. This makes ActSeq particularly valuable in scenarios where collecting large-scale datasets is impractical or costly. By enriching the training data with diverse and consistent augmentations, ActSeq provides an effective solution for overcoming data scarcity while maintaining high model performance.

