# OpenReview forum: "Action Sequence Augmentation for Action Anticipation"
_ICLR.cc/2025/Conference — ICLR 2025 Poster_

### Official Review · Reviewer_s45F · 2024-11-02

**Soundness:** 2
**Presentation:** 3
**Contribution:** 2
**Rating:** 6
**Confidence:** 5

**Summary:**

This paper introduces ActSeq, a data augmentation method for video action anticipation. To address the ordering bias in the training data, authors built a grammar base and alternate videos based on the grammar. Both observations and future action are alternated. The proposed method is evaluated on 50Salads, EGETA Gaze+, and EPIC-Kitchens-100. Results show ActSeq improves baseline methods consistently.

**Strengths:**

1) This paper tackels an important issue of action anticipation
2) The action goal is specifically modeled and can be interpreted
3) The proposed augmention method obtains promissing results on benchmark datasets.

**Weaknesses:**

1) For equation1, same atomic actions may appear multiple times in a video, the count may not be accurate.
2) One action may have more than one prerequisite actions, only modeling the pairwise dependencies may not be optimal since the high-order dependencies are not considered.
3) Some atomic actions have higher frequencies in training data, which leads may affect the AGI model
4) There are few comparison with other augmentations methods. Only random processing is compared in Table 6. Simple augmentation methods, such as rotation and adding random noise, may also improve the performance. To demonstrate the superiority of ActSeq, a comparison with other augmentation methods is preferred.
5) Details of generating augmented videos are missing

**Questions:**

1) How are the videos manimpulated? Do the augmented videos look realistic?
2) From the reulsts in Tables 3-5, why 5x augmentation is worse than 3x?

**Details Of Ethics Concerns:**

No concerns

---

> ### Author Response · Authors · 2024-11-22
> **Reply to s45F**
>
> We thank the reviewer for the feedback and for recognizing the importance of work.
>
> 1. For Equation 1, the same atomic actions may appear multiple times in a video; the count may not be accurate.
>
> - Thank you for pointing this out; you are correct that the same atomic actions may appear multiple times in a video. For example, in the sequence `[B → A → A → D → A]`, there are three instances of action B preceding action A. In this case, we only count `B → A` once in Equation 1, ignoring repetitive instances. This approach ensures a consistent and unbiased representation of action transitions.
>
> 2. One action may have more than one prerequisite action. Only modeling the pairwise dependencies may not be optimal since the high-order dependencies are not considered.
>
> - We agree that one action may have more than one prerequisite action, and modeling high-order dependencies is essential for effective action sequence learning.
>
> - To clarify, firstly, our grammar accounts for actions with multiple prerequisite actions. For example, “put apple into fridge” depends on the actions “take apple” and “open fridge.” Secondly, our grammar captures high-order dependencies through object interactions, which can be seen as "sub-goals" and are structured hierarchically. Object interactions are also actions (e.g., "put apple into fridge"). When the model learns pairwise dependencies, it learns not only the dependencies between atomic actions but also the dependencies between object interactions. These dependencies are considered high-order dependencies.
>
> 3. Some atomic actions have higher frequencies in training data, which may affect the AGI model.
>
> - We would like to clarify that when the grammar is induced based on pairwise action dependencies (as calculated using Equation 1), the frequency of individual atomic actions in the training data does not affect the AGI model. The model focuses on the **dependencies between actions** rather than their standalone frequencies, ensuring that the induction process remains robust and unbiased. We appreciate your observation and hope this explanation provides clarity.
>
> 4. Manipulation of videos and generation of augmented videos.
>
> - We would like to clarify that our augmentation approach does not augment visual features, manipulate videos or generate augmented videos, but focuses on the **ordering of actions**. Hence visual augmentations, such as rotation or adding random noise, are not directly comparable to our method, as they address a different aspect of the data. Generation of augmented videos is not within the scope of our paper.
>
> 5. From the results in Tables 3-5, why is 5x augmentation worse than 3x?
> - Only in Table 3, we observe that 5x augmentation is worse than 3x. The negative returns are obtained when augmenting the observed action sequence. This indicates a potential over-modification of the observed action sequences, where learning the next action from the augmented sequence becomes overly difficult.
>
> Thank you for your comments. We appreciate the opportunity to address these questions.

---

### Official Review · Reviewer_NV6R · 2024-11-04

**Soundness:** 3
**Presentation:** 2
**Contribution:** 2
**Rating:** 6
**Confidence:** 3

**Summary:**

Thanks the authors for this ICLR submission.

Summary:

This work identifies two major problem in action anticipation research: **1)** the default dataset split-up only allows for partial exploration over the entire action space, and **2)** noisey action labels induced by the imperfect preprocessing step (e.g., action recognition). To this end, this paper proposes a data augmentation method to enhance both the observable context and the prediction target (i.e., future steps). Moreover, a more noisy-robust version of data is generated by randomly deleting or replacing action elements.

**Strengths:**

- Easy to follow.
- Nice discussion on the three main issues in action anticiaption from line34-45!
- Approach is technical sound. Although, as mentioned in the future work, the optimal solution is to perfectly de-compose interwined action sequences, the method in this paper serves as a practical temporary solution.
- Interesting study on the order bias.
- Some Improvements over standard benchmarks across a variety of baseline methods.

**Weaknesses:**

- (Minor) The manuscript could be written better
    - Inconsistent dataset name in Table 3 & 4 and line 407 - EGTEA Gaze+, EGTEA Gaze, EGTEA. Please consider make the dataset names consistent throughout the submission, which will make it easier to follow and make connections.
    - Table 7, 8 & 9 captions are sentences, and should be ended with period. Please follow the basic academic writing protocol to better the  the wrtting.
    - Given no appendix, it is better to explain the evaluated **verb, noun, action** items for epic-kitchen. Please include a section explaining the evaluated items and/or terminology. It is recommended that academic paper should be self-contained, so that readers can find all information within a single paper.

- (Minor) The MAOB study is interesting, yet it is incomplete - none has been done to other experimented datasets (e.g., EGTEA and Epic-Kitchen). I would like to suggest extending such study to the EGTEA and Epic-Kitchen-100 datasets. It will be insightful to check the correlation between the reduction of order bias and the performance improvements, which will be a strong supporting evidence for th ActSeq. method to be accepted.
- (Minor) From Table 3 & 4, it is seen that the amount of useful information does not scale well the times (`X`) of augmentation, which is a concern. The solution to settle down the **X** hyperparameter seems critical. Please provide some discussion on the trade-offs here and initialize some considerations on how to decide this hyper-parameter.
- (Minor) This method requires really large number of augmentations, e.g., **12X** more sequences, to bring significant improvements (e.g., > 1 % in **Recall**). Arguably, it costs much longer training time and compute resources, specially on large datasets. The efficiency is questionable. Please provide information on the computational costs of their method compared to baseline approaches, or to discuss potential strategies for improving efficiency while maintaining performance gains.
- (Minor) It would be remiss to ignore similar ablation study as in Table 3 & 4 on the other two datasets, i.e., 50 Salads and Epic-Kitchen-100, as they differ from the EGTEA Gaze+ in size, domain and data distribution. Showing more information will be essential to better evaluate the generalizability of the ActSeq. method. Please consider complete the ablation section with such experiments.

Missing reference:
- On diverse asynchronous activity anticipation, ECCV  2020.

**Questions:**

see weakness.

---

> ### Author Response · Authors · 2024-11-22
> **Reply to #NV6R**
>
> Thanks to the reviewer for the thoughtful review and for acknowledging the strengths of our work.
>
> 1. The manuscript could be written better.
> - Thank you for your valuable feedback regarding the clarity and consistency of our manuscript. We have made the necessary amendments to address these points and improve the overall quality of the paper. Specifically:
>     - We have ensured consistency in the dataset names across the manuscript, standardizing all references to "**EGTEA Gaze+**."
>     - Captions for **Tables 7, 8, and 9** have been revised to follow proper academic writing conventions, with each caption ending in a period.
>     - We have included definitions of **verb, noun, and action** for in section 4.1 Dataset in our revised submission.
>
> 2. Extend MAOB study EGTEA Gaze+ and Epic-Kitchen-100.
> -  We have extended the MAOB (Mean Action Order Bias) study to include the **EGTEA Gaze+** and **Epic-Kitchens-100** datasets, as shown in the table below. For fair comparison, the augmented sequences generated by our ActSeq method are all 12X (3X for observed action and 4X for predicted action) of the original sequences.
>
>     | Dataset                             | 50Salads | EGTEA Gaze+  | Epic-Kitchens-100 |
>     |-------------------------------------|----------|--------|-------------------|
>     | **MAOB (original sequence)**        | 0.32     | 0.28   | 0.46              |
>     | **Original + augmented sequence by ActSeq (ours)** | 0.12     | 0.10   | 0.19              |
>
> - Our results demonstrate that ActSeq consistently reduces the MAOB across all three datasets. This reduction indicates that our augmentation method effectively mitigates action order biases, leading to a more balanced and representative dataset. Notably, the improvements in MAOB reduction aligns with observed performance gains. We have included the complete MAOB study in Appendix B of revised submission.
>
> 3. Trade-offs between performance and scale of augmented sequences.
>
> - We agree that the benefit of adding more augmented sequences diminishes as their number increases. This is due to the progressive reduction in ordering bias. This suggests that the initial few augmented sequences contribute the most significant gains.
> - Striking the right balance between performance and computational cost is essential when determining the optimal number of augmented sequences. Based on our experiments across the three datasets, with a MAOB of around 0.3 in the original sequences, setting the augmentation size to **6X** provides a practical starting point. This level effectively reduces ordering bias while maintaining computational efficiency. Beyond this point, the value can be adjusted, enabling an optimal trade-off between efficiency and the benefits of sequence augmentation.
>
> 4. Computational costs.
>
> - It is true that larger augmentation ratios require more compute resources. However, significant performance improvements can be achieved even with a lower augmentation ratio. For example, as shown in Table 5, using 2X augmentations results in a 1.7% improvement in Top-5 Recall, while 12X augmentations yield a 3.4% gain. This suggests that users can flexibly adjust the augmentation ratio based on their resource constraints and desired performance goals.
>
> - The runtime increases approximately linearly with the augmentation ratio, but slightly faster convergence is observed due to the diversity introduced by the augmented sequences. Potential strategies, such as prioritising sequences that have high impact in reducing biases, could help balance efficiency and performance.
>
> 5. Ablation study on 50Salads and Epic-Kitchens-100.
>
> - We have included additional ablation studies on the **50Salads** and **Epic-Kitchens-100** datasets in the appendix C to evaluate the generalizability of the ActSeq method across datasets with varying sizes, domains, and data distributions.
>
> - The results are summarized below. These results are consistent with our findings on EGTEA Gaze+ and demonstrate the adaptability of ActSeq to datasets of different scales and domains.
>    - **50Salads dataset:** We observe consistent improvements in Top-1 Recall with increasing augmentation ratios. A 2X augmentation for both observed and next actions achieves a gain of 0.7% over the baseline (53.2% → 53.9%), with the highest recall of 53.9% achieved at a 2X observed and 4X next configuration.
>    - **Epic-Kitchens-100 dataset:** The Top-5 Recall steadily improves with augmented sequences. A 2X augmentation increases recall by 1.2% (19.1% → 20.3%), and the highest performance of 20.8% is observed with a 3X observed and 4X next configuration.
>
>
>
>
> 6. Missing Reference: On diverse asynchronous activity anticipation, ECCV 2020.
>
> - Thank you for pointing out the missing reference. We have added the relevant citation in our related work.

---

> > ### Comment · Reviewer_NV6R · 2024-11-25
> > **Thanks and some more feedbacks**
> >
> > Thanks the authors for the response. Please see a few more feedbacks as follows.
> >
> > - (Minor) Table 5 of current manuscript, `EGTEAGaze+` -> `EGTEA Gaze+`. A space is missing.
> > - (Minor) On computation cost, another way to tackle the efficiency might be using the MAOB metric to weight the importance of data sequence during training - diminish the loss value of data that owns high order-bias values.
> > - (Minor) Could you bring more insights to an open question: **What is the upper / lower bounds of MAOB**? Imagine a real world scenario that some research team has collected a video dataset of goal-oriented activity procedures and is about to use ActSeq. How would they know the MAOB value of their dataset is out of a valid range (or within the valid range) and ActSeq is very much needed?
> > - (Minor) Would ActSeq corrupts a dataset without order bias?

---

> > > ### Author Response · Authors · 2024-11-25
> > >
> > > Thanks to the reviewer for the additional comments.
> > >
> > > 1. Missing space in Table 5.
> > > - Thank you for highlighting the missing space in "EGTEAGaze+". We have corrected it to "EGTEA Gaze+" in the revised manuscript.
> > >
> > > 2. Use of MAOB to improve training efficiency.
> > > - Thank you for the suggestion. We agree that MAOB metric, which is used to measure the bias, can be used to prioritise sequences that have high impact in reducing bias. This is an interesting direction, and we plan to investigate its potential in our future work.
> > >
> > > 3. Bounds and Real-world Application of MAOB.
> > > - The upper bound of MAOB is 1, representing maximum order bias, which occurs when each action pair appears in only one direction. For example for the action pair (A,B), only A → B (i.e. B follows A) appears in the dataset. The lower bound is 0, indicating no bias, which occurs when action pairs occur equally in both directions. For example for the action pair (A,B), A → B and B → A appear exactly the same number of times in the dataset.
> > >
> > > - We observed that reducing the MAOB to below 0.2 leads to diminishing returns in performance improvement. Therefore, 0.2 could serve as a practical threshold for a "bias-free" dataset.
> > >
> > > - Additionally, researchers can compare the MAOB values of the training and test sets to analyse the differences between the two. For example, if the training dataset has a high MAOB while the test dataset does not, then a model trained on the training dataset is likely to overfit to biased sequences, leading to poor generalisation on test data. It would be up to the researcher's decision either (i) to reduce the difference in order to ensure a consistent representation of action patterns during training and evaluation, or (ii) to retain the difference, in order to analyse the generalisation capability of a given model.
> > >
> > > 4. Impact of ActSeq on Datasets Without Order Bias.
> > > - If MAOB = 0, i.e. for action pair (A,B), both orders A → B and B → A occur equally in the original training dataset. Introducing an additional augmented sample (e.g., A → B) can temporarily create bias. However, adding complementary samples (e.g., B → A) restores balance.
> > > - To mitigate such risk, one can monitor pairwise MAOB changes before and after augmentation to ensure that ActSeq does not have undesirable impact.

---

> > > > ### Comment · Reviewer_NV6R · 2024-11-25
> > > >
> > > > Thanks for the additional response. Most of my concerns are addressed.
> > > >
> > > > To the reviewer, the theoretical derivatives of MAOB is the most valuable contribution of this work. The augmentations techniques are mostly practical engineer efforts and has been proven useful throughout experiments. Yet, what the reviewer is really looking for is a well-rounded solution building upon MAOB initiatives which can reduce the levels of heuristic designs through the paper, e.g.,  `X` times augmentations, and bring this method to broader use cases.
> > > >
> > > > Having seen all other reviews' comments, the reviewer would like to hold onto the score of 6.

---

### Official Review · Reviewer_VMPA · 2024-11-04

**Soundness:** 3
**Presentation:** 3
**Contribution:** 3
**Rating:** 8
**Confidence:** 4

**Summary:**

The authors introduce a data augmentation strategy, where they synthetically generate plausible, unobserved sequences of actions that obey the constraints of a grammar they induce over the action labels. They demonstrate this helps performance across a variety of models and datasets.

**Strengths:**

1) The induction of a grammar over actions is vitally important due to the combinatorial nature of action sequence modeling. Actions can be arranged quite flexibly, and video datasets are not large enough to span all the variations over action sequences.
2) The rules of the grammar capture well capture the hierarchical and compositional nature of action. This grammar they've induced can be applied in other domains (e.g. robotics, action recommenders, etc), which otherwise would normally be handwritten.
3) The models and datasets were chosen well.

**Weaknesses:**

1) The results for Tables 8 and 9 are reporting Top 5. Top 1 results for action anticipation predictions are also helpful to see. Your data augmentation approach likely best improves the Top 5 results, but could possibly be hurting the Top 1 results by training over plausible but unlikely future actions.
2) A qualitative figure would be nice to see. Or at least a visualization of the trees for one of the datasets.
3) I don't fully understand Table 1. Why are there negative values? There is no caption and the Figure is confusing. This figure is vital for understanding how Figure 3e) is constructed.

**Questions:**

1) Inducing a grammar over the observed actions is good, but I suspect an LLM could be used as an alternative source of action sequence augmentations, especially considering the Grammar operates only over narration-derived actions. Is there any reason to believe this isn't the case, or where one fails where the other would succeed? A qualitative example would be nice.
2) There's a paper (Therbligs for Action Understanding) [1] below that imposes a grammar over actions based on commonsense rules. They perform action anticipation as well, with improved performance due to these rules enabling the modeling of plausible sequences that abide by the rules. This paper should be included in the Related Works.
3) I wonder if this approach improves the data efficiency of a network, in that it needs less action sequences. This is claimed in Section 3 but not empirically explored (the way to do that would be to do an experiment where low-data model is compared with and without the augmentations).
4) Is there an augmentation strategy that rather than increase the augmentations by a set factor, uses the set of all plausible sequences for data augmentation?


[1] Dessalene, Eadom, et al. "Therbligs in action: Video understanding through motion primitives." Proceedings of the IEEE/CVF Conference on Computer Vision and Pattern Recognition. 2023.

---

> ### Author Response · Authors · 2024-11-22
> **Reply to #VMPA**
>
> Thanks to the reviewer for the thoughtful and detailed feedback, as well as for recognizing the strengths of our work.
>
> 1. Top 1 results for Epic-Kitchens-100 and 50Salads.
> - We specifically analyzed the top 1 performance as detailed in the tables below for EGTEA Gaze+ and Epic-Kitchens-100 datasets. The results indicate that our augmented sequences have minor but consistent improvement on top 1 accuracy on these datasets.
>
> ---
>
>      Top 1 Recall on EGTEA Gaze+
>     | Method                              | Action |
>     |-------------------------------------|--------|
>     | RULSTM (Furnari & Farinella, 2020)  | 19.5   |
>     | RULSTM + ActSeq                     | 19.8   |
>     | AVT (Girdhar & Grauman, 2021)       | 20.8   |
>     | AVT + ActSeq                        | 21.3   |
>     | RAFTformer (Girase et al., 2023)    | 21.2   |
>     | RAFTformer + ActSeq 66.3            | 22.1   |
>
> ---
>
>      Top 1 Recall on Epic-Kitchens-100
>     | Method                              | Action |
>     |-------------------------------------|--------|
>     | RULSTM (Furnari & Farinella, 2020)  | 4.7    |
>     | RULSTM + ActSeq                     | 4.9    |
>     | RAFTformer (Girase et al., 2023)    | 6.4    |
>     | RAFTformer + ActSeq                 | 6.6    |
>     | MotionFormer (Patrick et al., 2021) | 7.2    |
>     | MotionFormer + ActSeq               | 7.3    |
>
> - On deeper analysis into the correlation between bias and performance, we identify two scenarios where our augmentation strategy may improve or decrease the top1 performance. If the training and testing sets contain different biases (e.g. in training set, the sequence AB is more likely than sequence BA; in testing set, the opposite is true), addressing the biases in training set generally benefits the performance. Conversely, if both the training and testing sets contain the same biases, reducing the bias has no impact on the top1 performance. In all three datasets, both cases exist. The consistent improvement indicates that removing biases is beneficial for top 1 accuracy.
>
>
> 2. A qualitative figure or a visualization of the trees for one of the datasets.
>
> - In Appendix D of the revised submission, we have added a step by step demonstration of our augmentation method through an example from Epic-Kitchens-100 dataset, including a visualisation of the grammar tree.
>
>
> 3. Why are there negative values in Table 1?
>
> - Table 1 represents a skew-symmetric matrix that visualizes the dependencies between actions. In this matrix, **$M(i, j) = 1$** means Action $i$ depends on action $j$, and **$M(i, j) = -1$** means Action $j$ depends on action $i$, meaning $j$ typically follows or is contingent on the completion of $i$. We have added a more descriptive caption.
>
>
> 4. Comparison with LLM.
> - Yes LLMs could be used as an alternative source of action sequence augmentations. Grammar-based methods excel in interpretability with explicitly defined and traceable rules. LLMs, on the other hand, leverage vast datasets to generate diverse and scalable action sequences, making them more adaptable to less-structured environments. However LLMs are known to be prone to hallucinations.
>
> - In Appendix A of the revised submission, we compare a few representative LLMs, and our grammar-based method (ActSeq) on an example from the 50Salads dataset. LLMs have varied performance based on model capabilities, inherent biases, and occasional misunderstandings of task. While more sophisticated prompt engineering could improve LLM responses by guiding them towards more accurate predictions, it remains challenging to anticipate and cover all potential scenarios.
>
> 5. Paper to be included in related works.
>
> - Thank you for suggesting the inclusion of "Therbligs for Action Understanding" by Dessalene et al. This work has been added to our related works section in the revised manuscript.
>
> 6. Improving data efficiency.
> - In Appendix E, we show additional experiments with low-data models, which validate the data efficiency of ActSeq. Performance levels close to those of full-data models are achieved with fewer available original training samples. With 3 videos per task, our top 5 recall is 64.7%, compared to 63.5% of RAFTformer trained with full training set.
> 7. Use of all plausible sequences for data augmentation.
>
> - Actually, our parser has the capability to generate all plausible sequences for data augmentation. However, we observe that the performance gains diminish as biases are progressively reduced beyond a certain point. This is because further augmentation adds less incremental value once the model has already been exposed to a sufficiently diverse set of sequences. Moreover, using the full set of plausible sequences could impose significant computational costs, making it impractical in many scenarios. A balanced strategy that carefully selects a subset of plausible sequences can achieve an optimal trade-off between performance improvement and computational efficiency.

---

### Official Review · Reviewer_c22w · 2024-11-04

**Soundness:** 3
**Presentation:** 2
**Contribution:** 2
**Rating:** 6
**Confidence:** 3

**Summary:**

The paper introduces a data augmentation strategy (ActSeq) for the task of action anticipation. The paper recognizes multiple challenges of action anticipation - 1). multiple possible action sequences, 2). action ordering bias from the datasets, 3). interleaved actions from interleaved goals and present an augmentation strategy to address these challenges.

Given the action sequence video, video grammar is derived using Action Grammar induction algorithm. Then the cross-tree early parser predicts alternative next actions based on the observed actions and random modifications of insertion, deletion and replacement is further applied. Overall, this augmentation gives 3x (3 times) the size of the original dataset.

The method is evaluated on three action anticipation datasets - epic kitchens, egtea gaze+, and 50salads. The data augmentation strategy is incorporated with multiple prior works and their performance with and without augmentation is compared. These methods show improvement with data augmentation.

**Strengths:**

The paper addresses concrete challenges that arise in the task of action anticipation and presents a simple approach of data augmentation to address the ordering bias and multiple scenarios possibility when anticipating actions. The paper also presents an exhaustive evaluation on three action anticipation datasets. The idea of interleaved goals and interleaved actions is interesting.

**Weaknesses:**

The contribution and the novelty of the paper seem to be limited. Previous works [1,2] have explored some components of the proposed approach such as exploring grammar for activity prediction and data augmentation for temporal bias which is similar to the random modification of the proposed idea. Can the authors emphasize the contributions and especially the novelty of the work?


1. Piergiovanni, A. J., Anelia Angelova, Alexander Toshev, and Michael S. Ryoo. "Adversarial generative grammars for human activity prediction." In Computer Vision–ECCV 2020: 16th European Conference, Glasgow, UK, August 23–28, 2020, Proceedings, Part II 16, pp. 507-523. Springer International Publishing, 2020.
2. Mittal, Himangi, Nakul Agarwal, Shao-Yuan Lo, and Kwonjoon Lee. "Can't make an Omelette without Breaking some Eggs: Plausible Action Anticipation using Large Video-Language Models." In Proceedings of the IEEE/CVF Conference on Computer Vision and Pattern Recognition, pp. 18580-18590. 2024.

**Questions:**

The datasets explored in the paper predict only a single action given the observed actions. Would the proposed data augmentation strategy also work in scenarios where multiple actions (e.g. 20 actions) have to be predicted given the observed actions?

---

> ### Author Response · Authors · 2024-11-22
> **Reply to #c22w**
>
> We thank the reviewer for the valuable feedback and insightful suggestions. We appreciate the acknowledgment of our work’s strengths and the constructive feedback to improve our paper. Detailed responses addressing the concerns and clarifying our novelty and contribution are provided below.
>
> 1. Highlight the contribution and novelty compared to [1, 2].
> - **[1]:** Interleaving goals and actions result in numerous plausible action sequences, but action anticipation datasets often capture only a subset of these possibilities. This leads to biased action ordering, causing models to favor specific sequences and struggle to generalize to unseen or less frequent patterns. The work in [1], like in [3, 4, 5] (mentioned in related work), derives stochastic production rules directly from training data distribution, inheriting the ordering biases in the data. In contrast, our approach explicitly addresses these ordering biases by generating diverse plausible sequences, enriching the existing dataset and mitigating the biases.
>
> - **[2]:** While this work shares our goal of enhancing generative capabilities by producing alternative plausible sequences, it relies on **large language models (LLMs)** to generate these sequences. Our **grammar-based approach**, not only offers greater interpretability (e.g., clarity on decision-making and sequence generation) but also allows for direct manipulation and control of grammar rules, which is critical for applications requiring high precision, security, abidance of regulatory frameworks, etc. We have included a qualitative analysis of the augmented sequence generated by LLMs and ActSeq in appendix A. LLMs have varied performance based on model capabilities, inherent biases, and occasional misunderstandings of task. While more sophisticated prompt engineering could improve LLM responses by guiding them towards more accurate predictions, it remains challenging to anticipate and cover all potential scenarios.
> - We are grateful for highlighting the two relevant studies. We have included these references in our related work section.
>
> **References:**
> 1. Piergiovanni, A. J., et al. "Adversarial generative grammars for human activity prediction." *ECCV 2020*.
> 2. Mittal, H., et al. "Can't make an Omelette without Breaking some Eggs: Plausible Action Anticipation using Large Video-Language Models." *CVPR 2024*.
> 3. Pei, M., et al. "Parsing video events with goal inference and intent prediction." *ICCV 2011*.
> 4. Qi, S., et al. "A generalized Earley parser for human activity parsing and prediction." *TPAMI 2020*.
> 5. Si, Z., et al. "Unsupervised learning of event and-or grammar and semantics from video." *ICCV 2011*.
>
> ---
>
> 2. Would the proposed data augmentation strategy also work in scenarios where multiple actions (e.g., 20 actions) have to be predicted?
>
> - Yes, our data augmentation strategy can be adapted for **long-term action anticipation** by recursively predicting next actions. Using our augmented dataset directly, we evaluate the RAFTformer model, trained on short-term action anticipation, for long-term action anticipation task on Epic-Kitchens-100 dataset. The metric for evaluation was the **Edit Distance @ 20 Steps** (the lower, the better). The table below shows that our augmentation strategy improves RAFTformer's performance across actions, nouns, and verbs.
>
>     | Model                  | ED@20 Action | ED@20 Noun | ED@20 Verb |
>     |------------------------|--------------|------------|------------|
>     | RAFTformer (original)  | 96.4         | 75.2       | 74.7       |
>     | RAFTformer (w/ ActSeq) | 95.2         | 75.0       | 74.3       |
>
>
> - Alternatively we could generate new training samples for long-term action anticipation by allowing the grammar parser to forecast multiple steps ahead. The augmented samples can then be used to train long-term action anticipation models. This will be part of our future work.

---

> > ### Comment · Reviewer_c22w · 2024-11-25
> > **Acknowledgment**
> >
> > Thank you authors for thoroughly answering my concerns! Highlighting the differences related to mitigating bias and lesser interpretability of LLMs between the proposed grammar approach with the [1, 2] related works provided clarity. The approach can also be extended to long-term anticipation task which is a strength of the grammar-based approach and data augmentation strategy. I am inclined to raise my rating to 6 after reading the responses of the other reviewers.

---

### Meta-Review · Area_Chair_9F7z · 2024-12-20

**Metareview:**

This paper proposes a data-augmentation method for action anticipation based on action grammars.

The reviewers find the approach interesting, especially with respect to the interleaved goals and interleaved actions.

Some weaknesses raised include (a) clarifying distinctions between the work with respect to previous works that study action grammars and (b) additional baselines.  This has largely been addressed through the author response.

The AC has read through the reviewer comments and author responses. Given the strong support by all the reviewers (6,6,6,8), the AC recommends that the paper be accepted. The authors are requested to incorporate the content in their response to reviewers to their camera-ready version of the paper.

**Additional Comments On Reviewer Discussion:**

Clarifications on the method and experiments and additional experimental results came up during the discussion.  The authors are requested to add these to their camera-ready.

---

### Decision · Program_Chairs · 2025-01-22

Accept (Poster)